# Wine Faults: State of Knowledge in Reductive Aromas, Oxidation and Atypical Aging, Prevention, and Correction Methods

**DOI:** 10.3390/molecules27113535

**Published:** 2022-05-31

**Authors:** Štefan Ailer, Silvia Jakabová, Lucia Benešová, Violeta Ivanova-Petropulos

**Affiliations:** 1Institute of Horticulture, Faculty of Horticulture and Landscape Engineering, Slovak University of Agriculture in Nitra, 94976 Nitra, Slovakia; stefan.ailer@uniag.sk; 2Institute of Food Sciences, Faculty of Biotechnology and Food Sciences, Slovak University of Agriculture in Nitra, Tr. A. Hlinku 2, 94976 Nitra, Slovakia; xbenesova@uniag.sk; 3Faculty of Agriculture, University “Goce Delčev”-Štip, Krste Misirkov 10-A, 2000 Štip, North Macedonia; violeta.ivanova@ugd.edu.mk

**Keywords:** wine faults, reductive aromas, browning, atypical aging, preventive measures, corrective solutions

## Abstract

The review summarizes the latest scientific findings and recommendations for the prevention of three very common wine faults of non-microbial origin. The first group, presented by the reductive aromas, is caused mainly by excessive H_2_S and other volatile sulfur compounds with a negative impact on wine quality. The most efficient prevention of undesirable reductive aromas in wine lies in creating optimal conditions for yeast and controlling the chemistry of sulfur compounds, and the pros and cons of correction methods are discussed. The second is browning which is associated especially with the enzymatic and non-enzymatic reaction of polyphenols and the prevention of this fault is connected with decreasing the polyphenol content in must, lowering oxygen access during handling, the use of antioxidants, and correction stands for the use of fining agents. The third fault, atypical aging, mostly occurs in the agrotechnics of the entire green land cover in the vineyard and the associated stress from lack of nutrients and moisture. Typical fox tones, naphthalene, or wet towel off-odors, especially in white wines are possible to prevent by proper moisture and grassland cover and alternating greenery combined with harmonious nutrition, while the correction is possible only partially with an application of fresh yeast. With the current knowledge, the mistakes in wines of non-microbial origin can be reliably prevented. Prevention is essential because corrective solutions for the faults are difficult and never perfect.

## 1. Introduction

We are currently witnessing a sensory wine revolution. There are liberal styles and fractions, where oxidation, turbidity, or excessive content of phenolic substances are accepted even in white wine. Protein-dependent turbidity and crystalline sediments do not need to be classified as wine faults in the current liberal conditions [1,2]. If there is a market for such wines, and they are produced according to clear applicable rules, it is necessary to respect them. Despite this liberal era, the two-thousand-year history has set and shaped certain rules. It is documented and published in professional and scientific literature, what can be considered a faulty wine with the impacts on the wine sensory profile [3,4,5,6,7]. Consuming faulty wine does not bring pleasure to the consumer, it causes unpleasant feelings, and the consumer does not ask for another sip [4,8].

Wine faults in traditional wine-growing countries are divided into those that are not caused by the activity of microorganisms, which can be relatively reliably corrected (oxidation, atypical aging, reductive aromas, various odor disorders, non-harmonic ratio of components), and faults caused by microbial activity with consequencesthat cannot be completely remedied (vitrification, brett, refermentation, mouse taint, mannitic fermentation, undesired decomposition of acids) [4,9,10,11].

Wine faults of non-microbial origin are caused either by various physical and chemical processes in the vineyard, the wine, or by contamination from the environment. Under these influences, wine changes its appearance, aroma, and taste. The most effective way to prevent the faulty wine in stock is good agricultural technology and the precision and consistency of professional staff [4,11].

The precursors for wine faults are physical, chemical, or microbiological, arising from incorrect agricultural technology, low ethanol content, high acid content, high content of phenolic substances (in white and rosé wines), and incorrect sulfurization regime, improper storage, and oxygen regime [4,10,12,13].

The objective of this review was to contribute to the current knowledge of common wine faults related to reductive aromas, browning, and atypical aging to summarize the latest knowledge on these topics, particularly in terms of sensory attributes, chemical background, and the preventive and corrective measures for wines during fermentation and after bottling. The review contributes to the knowledge of the potential for managing the faults during winemaking and wine storage.

## 2. Reductive Faults

Sulfur, as an important element in biological systems, is metabolized in various compounds and is responsible for the desired but also sometimes unpleasant aromas [14,15,16]. The complex chemical changes during wine fermentation include sulfur-related chemical pathways connected with wine-yeast metabolism. Along with the desirable volatile sulfur compounds that are perceived positively, such as 3-mercaptohexan-1-ol, 3-mercaptohexyl acetate, 4-mercapto-4-methylpentan-2-one, and 4-mercapto-4-methylpentan-2-ol with citrus, grapefruit and passionfruit tones in wine aromas, other volatile sulfur compounds contributing to so-called reductive aromas are likely generated through the same winemaking techniques [15,17]. Most of these undesirable volatile sulfur compounds are produced by yeast in sulfate assimilatory and dissimilatory reduction pathways [15,18,19] but H_2_S and mercaptans can also be formed via an alternative biochemical route from the other sources such as glutathione or sulfane sulfur compounds which are not well elucidated [14,16,20].

Despite the implementation of good winemaking practices, it is not uncommon for reductive aromas to appear in wine during vinification [21]. Goode and Harrop [22] reported that reductive faults are responsible for 30% of all faults in commercial wines and should be considered seriously due to the significant economic impact on winemakers. The discernment of undesirable sulfur off-aromas due to higher concentrations of the above-mentioned compounds in finished wine has a negative effect and devalues a wine’s quality, and is a possible reason for the rejection of a faulty wine by consumers. Volatile sulfur compounds thus present a challenge for modern-day winemaking. It is desired to limit (or eliminate) the production of undesirable H_2_S and thiols but at the same time, maintain and enhance the production of the favorable volatile thiols [18].

### 2.1. Sensory Attributes of Reductive Faults

Besides the desired S-containing compounds that are important for wine quality, some substances, even in very low concentrations, cause off-odors and strong undesired aromas [15,23]. Sensory attributes associated with reductive aromas are rotten egg, putrefaction, sewage-like, rotten cabbage, onion, and burnt rubber. The most important volatile sulfur compounds (VSCs) linked with these descriptors are hydrogen sulfide (H_2_S), methanethiol (MeSH), and ethanethiol (EtSH) [24]. Other S-containing compounds with a negative effect on the sensory properties of wine are dimethylsulfide and benzenemethanethiol. Their impact on wine aroma and flavor is due to their high volatility, reactivity, and low threshold concentrations (Table 1). These compounds are well known and belong to the commonly occurring problems in winemaking [15,16,25] and present an interesting topic.

The contribution of organic and inorganic sulfur-containing compounds to aroma characteristics and their impact on wine aroma perception are associated with problems or wine faults if they are present in concentrations higher than their odor threshold [15,24] (Table 1).

### 2.2. Biochemical Background of Volatile Sulfur Formation

Sulfur can be present in eight oxidation states from S (−II) up to S (+VI) [16]. Due to this, sulfur can participate in an array of oxidation, reduction, and disproportionation reactions that are inorganic, and, in addition, these reactions are connected with microbial metabolism. In the process of fermentation, pesticide residues on grapes containing elemental sulfur are not metabolized only to hydrogen sulfide, but it is possible to form precursors that generate H_2_S in the post-fermentation stage after bottling. The known precursors are glutathione tri- and polysulfanes (Glu-S-Sn-S-Glu) and recently also tetrathionate (S_4_O_6_^2−^) was identified [16].

The wine fermentation process is supported by wine yeast and involves changes in the composition of sulfur compounds that are connected with the biosynthesis of S-containing amino acids cysteine and methionine through the sulfate assimilatory reduction pathway [16,18,30,31,32,33].

The chemical and microbiological transformations of sulfur-containing substances are accompanied by sensorial changes and not all of them are desirable in the final product [16,24]. Wine, accompanied by the smell of rotten eggs, indicates the presence of hydrogen sulfide (H_2_S), and its derivatives (methanethiol, ethanethiol, etc.) which are products of yeast metabolism during fermentation and other transformations during the post-bottling stage [16,24].

#### 2.2.1. Formation of Hydrogen Sulfide, Methanethiol, and Ethanethiol

Explaining the formation of H_2_S in the fermentation process of grapes has been a research topic for many authors, e.g., Rauhut [34], Swiegers and Pretorius [18], Ugliano and Henschke [35], Ugliano et al. [36], Cordente et al. [37], etc. The presence and concentration levels of S-containing organic compounds such as aminoacids and peptides (cysteine, methionine, S-adenosylmethionine, and glutathione) are essential for the wine yeast metabolism and growth. An insufficient concentration of these compounds in the yeast diet leads to their synthesis from inorganic sulfur sources by the yeast cells. The first step of this synthesis is the reduction of inorganic ions of sulfites and sulfates to hydrogen sulfide, which is a precursor of sulfur-containing amino acids. The formation of hydrogen sulfide is present in every fermentation process, but its further use in metabolic pathways of yeast is closely dependent on the levels of other substrates—nitrogenous substances that are essential in the formation of S-containing aminoacids and their peptides and proteins. Excessive H_2_S production by yeast is affected by the three main factors during wine fermentation: assimilable nitrogen, sulfur dioxide, and yeast strain [38].

This process of H_2_S formation is present in every wine, however, its levels differ based on the above-mentioned conditions. The lack of nitrogenous substances in the yeast diet is a trigger for excessive production. This explanation was supported by the studies of Ugliano et al. [36] and Müller et al. [16] who reported that sulfur sources, such as fungicide residues containing elemental sulfur, hydrogen sulfide, and methanethiol, etc., that are formed by yeasts are considered to be the main substrates for the generation of latent precursors of off-odor compounds during the vinification. Ferrer-Gallego et al. [39] mentioned that sulfur dioxide is connected with the formation of hydrogen sulfide via yeast metabolism.

According to several studies [32,36], the lack of nitrogenous substances in the must is considered the main reason for hydrogen sulfide formation. On the other hand, some experiments proved that H_2_S is also formed in conditions where assimilable nitrogen is present or supplemented [12,32,36]. Ugliano et al. [12] stated that nitrogen supplementation in wine to influence the formation of H_2_S stem is from the initial yeast assimilable nitrogen and yeast properties that produce H_2_S. The explanation of the excessive formation of H_2_S is based on the permanent exposition of vineyards to stress conditions (drought, malnutrition, overload).

The biochemical pathway for the formation of hydrogen sulfide can be described through the wine yeast metabolism (Figure 1). Inorganic (sulfate, sulfite, sulfur dioxide), as well as organic sulfur compounds (cysteine and glutathionine), are reactants for the formation of H_2_S [18,38]. The biochemical pathway in *S. cerevisiae* was elucidated by Yamagata [40] and Rauhut [34] as a sulfate reduction sequence (SRS) pathway. Current knowledge of the regulatory mechanisms are described in the work of Guidi et al. [38] and Müller et al. [16].

Inorganic sulfur sources are normally present in the grape must, but the must does not contain organic-binding sulfur which leads to the biosynthesis of sulfur compounds that are important for wine yeast [41,42,43]. Basically, H_2_S is formed either from the HS^-^ ions which present a metabolic intermediate from sulfate and sulfite reduction that are important for the synthesis of organic sulfur compounds. In this step of metabolism, nitrogen supply is essential for further reactions in which HS^-^ ions are transformed especially to the S-aminoacids methionine and cysteine through the O-acetylserine and O-acetylhomoserine, which are formed during nitrogen metabolism [41,42,43]. A lack of nitrogen sources or its unsuitability causes an end to the biochemical reaction with the formation of hydrogen sulfide, which first accumulates in the yeast cell and then diffuses through the membrane into fermenting must [19,30,44,45,46].

The sulfur reduction pathway, especially in longer maturation, allows the conversion of cysteine, methionine, and glutathione in H_2_S [47]. However, in the growth phase of yeast-free S-amino acids, they are bound to the proteins and as the fermentation proceeds, these free molecules can be released from yeast into the finished wine. Degradation of glutathione into amino acids happens in the conditions of cellular nitrogen deficiency [48], and the deficiency of nitrogen can also lead to the release of H_2_S from cysteine [18].

The role of genetic-based ability in the formation of H_2_S and other volatile sulfur compounds was proved in different yeast strains [18,39,49,50].

Besides the hydrogen sulfide, yeasts also assimilate through the sulfur metabolism of MeSH, EtSH, etc., [18,48]. Sulfhydryls, methanethiol, and ethanethiol are also responsible for sulfurous unpleasant off-odors due to their low odor thresholds and suppression of fruity and floral wine aromas [24,51].

The formation of methanethiol is possible from methionine via transamination and activity of demethiolase [18,31,51]. Methanethiol is able to be esterified to metylthioacetate [23,52]. Ethanethiol was explained to be formed in vitro in the reaction of hydrogen sulfide with ethanol or acetaldehyde [50].

In the post-bottling stage, hydrogen sulfide levels in wine are influenced by several determinants [17]. Their formation in the post-bottling stage is associated with the possible decomposition of cysteine followed by the accumulation of hydrogen sulfide in wine [53]. An important role is probably played by the concentration of oxygen after fermentation and during storage. A lower concentration affects the higher generation of hydrogen sulfide during post-bottling [17,54,55]. Nguyen et al. [56] investigated the influence of micro oxidation on Cabernet Sauvignon wine in the oxygen doses of 5–20 mg L^−1^ for one month in combination with malolactic fermentation. They observed some decreases in the levels of sulfur off-odors.

Sulfhydryl compounds are able to react with wine components, and the reductive components can be captured in the form of precursors. During the wine storage, decomposition of these precursors may occur resulting in the release of malodorous compounds. The precursors can be additionally induced by wine treatments (e.g., aeration, copper fining), which are performed to avoid or decrease reductive aromas in the wine before bottling [16].

Factors that impact the formation of H_2_S are as follows: the presence of a higher concentration of elemental sulfur, sulfurization with sulfur dioxide, the presence of compounds with organically bonded sulfur, a deficiency of pantothenic acid, and the existence of amino acids [18,31,57].

#### 2.2.2. Post-Fermentation Reductive Aromas—Thiols and Disulfides

The odor of hydrogen sulfide spreads from yeast from bottom to top. Post-fermentation generation of methanethiol and hydrogen sulfide has been explained based on several hypotheses including non-enzymatic reactions. The formation of methanethiol and ethanethiol has been proposed by the non-enzymatic reduction of symmetrical disulfides, thioacetate, and thioether hydrolysis, and decomposition of S-amino acids [58].

Therefore, the fermenting wine from the lower layers of the container and the yeast sludge must be checked regularly, at least once every three days. If hydrogen sulfide is not removed from the wine in time, more complex sulfur compounds ethanethiol and disulfides are formed that create a strong odor of rotten onions and feces of various intensities (mercaptans, post-fermentation sulfur taint). In the study by Kreitman et al. [59], in anoxic conditions during storage, the concentration of both hydrogen sulfide and methanethiol increased. Application of Cu(II) to the wine in amounts over 2-fold molar higher than the concentration of volatile sulfur compounds (H_2_S, MeSH, EtSH), resulted in the complete removal of all sulfhydryls [58]. In the same study, the addition of tris(2-carboxyethyl) phosphine was poorly efficient in releasing hydrogen sulfide from its copper complex. The efficiency of other copper chelators was studied in order to release sulfhydryls from their complexes, e.g., bathocuproinedisulfonic acid [58].

Ethanethiol can be removed from wine relatively reliably by applying copper salts. In the post-bottling study of Bekker et al. [55], the impact of Cu^2+^ in combination with sulfur dioxide (SO_2_) was followed in relation to hydrogen sulfide (H_2_S) formation in Shiraz and Verdelho wines. Treatment with copper sulfate in combination with oxygen exposure and glutathione in the post-bottling study on hydrogen sulfide and methanethiol was studied by Ugliano et al. [54]. Copper addition resulted in H_2_S accumulation during the second 3 months of storage and its highest concentrations were observed, especially in variants with glutathione and copper treatment in low-oxygen conditions. However, copper fining is generally considered as a manner of sulfhydryl removal [60,61], in conditions with a low-oxygen exposure which seems to promote the increase of H_2_S concentration in wine in the post-bottling stage [54,59].

According to Franco-Luesma and Ferreira [51], both de novo formation of H_2_S from precursor compounds as well as the release of H_2_S from metal complexes contribute to the final concentration of H_2_S formation in wines post-bottling, with the release from metal complexes responsible for the majority of H_2_S produced in red wines, and de novo formation responsible for the majority of H_2_S produced in white wines and rosé wines. Additionally, Franco-Luesma [62] suggests that the release of free H_2_S and MeSH from bound sources is a function of a decrease in the redox potential of wines.

Disulfides do not react or react only very weakly with copper salts, moreover, they are precursors of further ethyl mercaptan formation. Disulfides can be partially removed from wine by adsorbents, the best of which have the ability to bind odors. Activated carbon, as the main component of adsorbent was studied by Huang et al. [63] and was proved as efficient in the elimination of three types of disulfides (diethyl disulfide, dimethyl disulfide, and dimethyl trisulfide). It is a drastic intervention when besides the off-odors bouquet substances, other valuable substances are removed from the wine. Therefore, analysis of sulfur-containing compounds is important for wine quality control and research [16].

### 2.3. Preventive Measures of Formation Reductive Aromas

The most probable reason for the formation of sulfur-related flaws is a combination of circumstances: strong sulfurization of must or mash before fermentation, deficiency of non-essential aminoacids [36], and incorrect application of nitrogenous substances (nutrition) during fermentation. The best prevention of its occurrence is, therefore, the prevention and elimination of these factors [64].

The treatment of containers with sulfur dioxide vapors has to be ensured with slow-burning sulfur slices. The slices need to be ignited from above. If we ignite the slice from the bottom part, due to the generation of a large amount of heat during rapid combustion, a part of the elemental sulfur does not burn, rather melts and drips to the bottom of the vessel. Such unburned sulfur can later become the source of sulfur for the formation of H_2_S by yeast. Sulfur slices must therefore burn slowly and elemental sulfur must be completely oxidized into sulfur dioxide [64].

Several approaches in the elimination of undesirable off-odors from hydrogen sulfide were performed and proposed: (1) macro-oxygenation and aeration [21,55,65,66,67], (2) yeast assimilable nitrogen (monitoring, supplementation in wine) [15,21,36,68,69], (3) application of fining agents (especially copper fining) [21,58,70,71], and (4) application of lees [21,72].

The classical approach to decreasing the content of H_2_S in wines is aeration [65,66]. This method allows hydrogen sulfide removal with the immediate bottling of wine from lees with air and its continuous sulfurization (in a dose of 20 mg SO_2_ L^−1^). This procedure involves three processes: oxidation of hydrogen sulfide with atmospheric oxygen, venting of the hydrogen sulfide in the gas phase, and its partial reaction with sulfur dioxide, to form elemental sulfur. Aeration is successful in removing sulfhydryls as well as thioacetates [55]. However, simple aeration is not so effective for organic sulfides such as methanethiol and ethanethiol or heavier mercaptans. Anoxic conditions during the wine storage contribute to the permanent or increasing presence of these compounds resulting in the reductive character of the finished wine [67]. Targeted oxygenation also has beneficial effects on a healthy, smooth fermentation process, thus limiting the formation of hydrogen sulfide. The oxygenation must be done at temperatures below 16 °C. Enrichment of must with oxygen during oxygenation plays an important role in the formation of sterols and unsaturated fatty acids, which are important factors for the harmonious multiplication of yeasts and the integrity of their cell walls. In white wines made from oxidized must, hydrogen sulfide is formed in excessive quantities only rarely [73].

Remediation strategies, such as the addition of diammonium phosphate during fermentation, copper fining, the addition of fresh lees or lees products to wine, and aeration of the must during and after fermentation, are commonly employed in an effort to prevent the formation or to remove undesirable volatile sulfur compounds [21].

### 2.4. Corrective Solutions in Reductive Aromas

If the hydrogen sulfide odor in the wine is very strong (which means it was detected late), simple aeration and clarification may not be sufficient. Remediation is usually based on the selectivity of certain methods directed toward the sulfur compounds responsible for the reductive faults.

Certain treatments are selective in their ability to remove different types of sulfur species (i.e., sulfhydryls, disulfides, thioacetates, dialkyl sulfides), while other treatments may have associated risks [55]. Removal of hydrogen sulfide and methanethiol was attempted by the application of reducing agents, metal chelators, and maintaining lower levels of oxygen in red and white wines. Application of copper salts (sulfate or citrate) is a common way of this treatment [16] and was studied by many researchers (e.g., Smith et al. [15]; Bekker et al. [40]; Kreitman et al. [58]; Kreitman et al. [59]; Allison [66], etc.) Th addition of copper salts after fermentation results in the formation of non-volatile complexes and anoxic conditions in combination with other factors that can release sulfhydryls with an impact on promoting the off-odors. Treatment of finished wines with copper sulfate is due to the formation of stable complexes with sulfur compounds in order to eliminate the generation of H_2_S and methanethiol [15]. The precipitation of H_2_S with copper ions produces insoluble copper sulfide, which is an odorless compound. This will remove the hydrogen sulfide from the wine without aeration.

The use of copper sulfate poses an interesting dilemma to the winemaker as it is used to treat wines tainted with H_2_S and mercaptans, but at the same time, it reduces the concentration of the desirable volatile thiols as the Cu^2+^ ion does not discriminate between the two classes of sulfur compounds [18]. Copper fining is also not efficient for sulfides, disulfides, and thioacetates, and is limited to H_2_S and thiol removal [24]. The formation of disulfides and trisulfides contributing to additional off-odors due to copper fining, as well as problems associated with wine instability and additional wine-processing logistics, should also be taken into account as an unwanted impact of copper fining [24].

A comparative study was performed by Bekker et al. [21] in order to propose the most efficient remediation technique for maintaining fruit attributes and eliminating the reductive tones. In the study on the Shiraz wine variety, the fermentation process was combined with the use of different chemical and microbiological approaches. Techniques based on the application of lees, diammonium phosphate, and copper resulted in a fall in the fruit character of wines, and moreover, enhanced reductive aromas. A commonly described adverse effect of copper fining lies in the risk of copper reactivity with thiols in wine resulting in loss of the varietal character of wine [12]. Moreover, the elimination of copper sulfide from treated wine is sometimes needed. Its removal has been studied with the use of bentonite [71].

However, even with the chemical removal of the sulfur taint, it is necessary for the treated wine to be separated from the yeast sludge by bottling. The dose of the copper preparation to the wine must be determined in the laboratory conditions so that we do not unnecessarily burden the wine. Copper is a heavy metal and therefore it is always necessary to consider whether its use is really necessary. Especially winemakers without experience and laboratory equipment may have copper residues in the wine and then it is questionable whether this wine serves the health of consumers. When using copper preparations, it must be kept in mind that the statutory maximum permissible copper value in wine of 1 mg Cu^2+^ L^−1^ must be complied with for health safety reasons (EC Commission Regulation 606/2009) [74]). In practice, a maximum value of 0.6 mg L^−1^ is calculated. When calculating the dose of copper preparations, it should be kept in mind that the wine also contains a certain amount of copper, which comes from preparations for the treatment of grapes against fungal diseases. In a three-year survey, Ailer [64] found values of 0.06 to 0.61 mg Cu^2+^ L^−1^ in Pinot Blanc and St. Laurent varieties.

## 3. Wine Browning (Oxidation)

Wine oxidation is one of the major problems encountered in winemaking, and occurs in red, rose, and is especially evident in white wines. Browning is a quality defect in wine that is the result of a complex series of oxidation reactions that take place during processing, aging, and storage, which give rise to a brown color that increases color intensity, decreases brightness, and raises the browning index [75,76].

### 3.1. Sensory Attributes of Wine Browning

Wine browning is a result of complex non-enzymatic, oxidative, and also enzymatic changes, especially in white wines. Polyphenols in the white and red wines contribute to wine quality and influence further reactions leading to the development of not only desirable sensorial properties.

They play a key role in browning which has a significant effect on wine’s organoleptic characters and antioxidant properties. The result of the series of reactions increases the color intensity, decreases the brightness index, and raises the browning index. The change of wine color is characterized as a gradual replacement of the initial pale-yellow color to a brown-yellow color developed from access to oxygen [77]. Besides the color alteration, non-enzymatic oxidation may result in the appearance of a smell of rotten fruit, wood, or cooked vegetables [76,78]. As a result of the browning process, there is also a loss of varietal aroma and flavor, and a development of bitterness and astringency that can cause serious changes in the physical appearance and aroma properties of wine [76,79,80,81].

### 3.2. Biochemical Background of Wine Browning

From a sensory perspective, controlled oxidation could be beneficial for red wine by enhancing and stabilizing color and reducing astringency, however, the quality of white wine is generally damaged by excessive exposure to air [73,82,83]. Wine phenolics belong to two main groups: flavonoids (anthocyanins, flavan-3-ols, flavonols, and dihydroflavonols) and nonflavonoids (hydroxybenzoic and hydroxycinnamic acids and derivatives, stilbenes, and volatile phenols). The most common flavonoids are flavan-3-ols and flavonols (and anthocyanins in red wines) (Figure 2). Flavan-3-ols, mainly catechins and catechin–gallate polymers, produce a number of oxidation products that could be regarded as browning agents in white wines [84]. Regarding the group of flavonols, quercetin derivatives are the main components present in white wines. The major nonflavonoid phenolic compounds in white wines are hydroxycinnamic acid derivatives (Figure 3) which are easily oxidized components contributing to the browning of wines during aging.

The grape phenolic composition is affected by several factors such as grape variety, ripening stage, climate, soil, place of growing, and vine cultivation. Winemaking technologies, such as maceration time, temperature, the intensity of pressing, yeast, and SO_2_-doses, together with enological practices and aging, also modify the phenolic content in wine [85,86]. Usually, white wines are made without aeration in order to avoid extensive contact with oxygen, preventing browning of the wine and deterioration of the overall quality. Controlled low temperatures (14–18 °C) are usually applied for white wine production.

Then, the phenolic substances present, although not volatile in themselves, catalyze the decomposition of the aromatic profile of the wine and the oxidation processes. In the oxidation processes in wine, quinones and peroxides are formed in the presence of phenolic substances. The result is wine browning and an oxidized flavor. In the production of red wines, the mash is macerated throughout the fermentation. For this reason, the content of polyphenols (mainly anthocyanins) in red wines is higher, 1–5 g per liter, compared to white wines, which contain 0.2–0.5 g of polyphenols per liter [87].

#### Enzymatic and Nonenzymatic Oxidation

During the winemaking process, oxygen is an important factor in creating quality wines as it takes part in enzymatic and nonenzymatic oxidation reactions. Enzymatic oxidation almost entirely occurs in the grape must, while non-enzymatic reactions can happen both in grape must and wine [82,88,89].

Enzymatic oxidation is mediated by two distinct polyphenol oxidases which give rise to the formation of quinones (brown compounds): (i) A catechol oxidase, derived from healthy grapes, has both cresolase activity (hydroxylation of monophenols to ortho-diphenols) and catecholase activity (oxidation of ortho-diphenols to ortho-quinones), and; (ii) A laccase of fungal origin that does not have cresolase activity but does catalyze the oxidation of several types of phenols, in particular para-diphenols [82]. Enzymatic oxidation of selected phenolic acids in wine is shown in Figure 4.

During the non-enzymatic oxidation process, also called chemical oxidation of wine, the oxidative process is favored by the oxidation of polyphenols containing an ortho-dihydroxybenzene moiety (a catechol ring) or a 1,2,3-trihydroxybenzene moiety (a galloyl group), such as (+)-catechin/(−)-epicatechin, gallocatechin, gallic acid, and its esters, and caffeic acid, which are the most readily oxidized wine constituents [82]. These substrates are sequentially oxidized to semiquinone radicals and benzoquinones while oxygen is reduced to hydrogen peroxide. The whole process is mediated by the redox cycle of Fe^3+^/Fe^2+^ and Cu^2+^/Cu+ [90]. Other compounds with more isolated phenolic groups such as malvidin, p-coumaric acid, and resveratrol are oxidized at higher potentials.

Caftaric acid (caffeoyl tartaric acid) is the most abundant phenol in must. The polyphenol oxidases have a high affinity for this acid, so the corresponding ortho-quinone is the main enzymatic oxidation product of must. In addition to the high concentration of the quinone, the caftaric acid quinone/caftaric acid redox couple has a high redox potential. This molecule is therefore highly reactive, meaning that it participates in other redox reactions such as the oxidation of ascorbic acid and sulfites, and also of orthodiphenols that are not substrates of polyphenol oxidases. Another less abundant but not less important quinone is p-coumaryl tartaric acid, which is derived from the oxidation of coumaryl tartaric acid (coutaric acid) [91].

**Figure 4 molecules-27-03535-f004:**
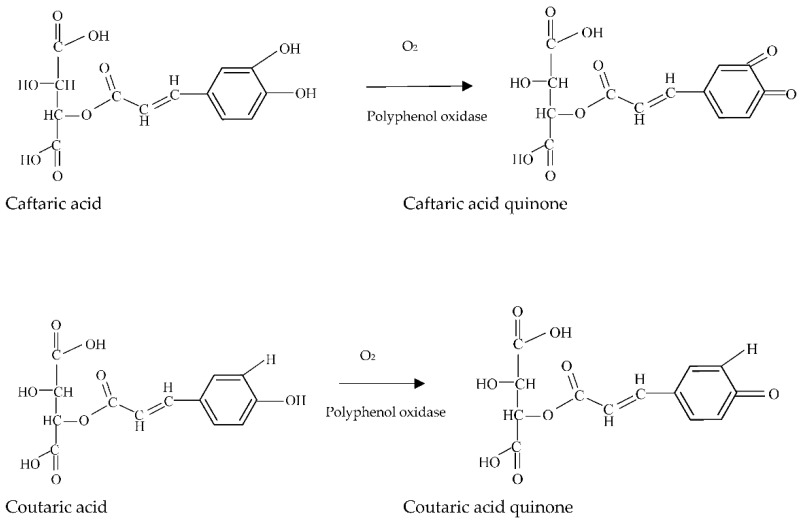
Enzymatic oxidation reactions of caftaric and coutaric acids in must and wine, based on Moreno and Peinado [91].

The enzyme polyphenol oxidase (PPO) reacts with naturally present phenolic compounds and catalyzes their oxidation into highly reactive primary quinones. The phenol/quinone redox couple has a high redox potential, meaning that quinone tends to be reduced, leading to the oxidation of other phenolic compounds, notably oligomeric flavanols and anthocyanins. This phenomenon is known as coupled oxidation. Quinones formed in this manner are called secondary quinones. They are highly unstable and give rise to condensation products (with their reduced form or with caftaric acid) [92].

Several compounds in must and wine have natural antioxidant activity and react with primary quinones, reducing their oxidizing action. One of these compounds is glutathione which is an important tripeptide in must and is highly abundant in certain grape varieties (Figure 5). Glutathione reacts with the quinone of caftaric acid and forms the acid 2-S-glutathionyl caftaric acid, also known as grape reaction product (GRP). The formation of GRP, which is colorless and does not provide a substrate for polyphenol oxidase, paralyzes the formation of brown products, as it inactivates the quinone of caftaric acid and therefore does not give rise to coupled oxidation reactions. GRP, however, can be oxidized in the presence of an excessively high concentration of caftaric quinones once glutathione is depleted, giving rise to intense browning. The browning of grapes must therefore depend on the relative proportions of glutathione and hydroxycinnamic acids [93,94,95].

### 3.3. Preventive Measures of Wine Browning

Wine is a highly complex matrix containing many potentially oxidizable compounds, including phenolic compounds, certain metals, tyrosine, and aldehyde, of which the flavonoids (e.g., dihydroxyphenolic compounds) and nonflavonoids (e.g., caffeic acid) can lead to brown products [92,96,97]. Therefore, during storage and treatment during cellar handling, it is necessary to prevent the access of atmospheric oxygen to the wine. With the reducing environment and convenient sulfurization regime, it is possible to prevent wine browning and thus reduce the use of antioxidants, especially sulfur dioxide. Since the main compounds that cause oxidation are phenolics, the intensity of browning (oxidation) depends on the content of phenolic compounds, the degree of wine exposure to atmospheric oxygen, and the degree of neglect of sulfurization [73,98].

Prevention of browning is possible by proper management of phenolic substances during the grape harvesting, pressing of grapes, the subsequent processing of must, regular filling up of the containers, application of a shielding gas atmosphere, and sulfurization. Susceptibility of wine to browning is possible to determine by the analysis of the polyphenolic content, or by using an accelerated test for browning capacity [75].

In anaerobic conditions, redox systems are in equilibrium at certain time intervals. Any supply of oxygen to the wine immediately disturbs the equilibrium state. Oxygen diffuses into musts or wines and reacts with easily oxidizable compounds and oxidizes them. In these reactions, peroxides can be formed that affect other oxidation and radical processes in the wine. There are always iron and copper cations in the wine, which break down the peroxides into the water and active oxygen. The released active oxygen also oxidizes those compounds which are not oxidizable by the molecular oxygen O_2_. Compounds such as L-ascorbic acid, oxoacids, amino acids, and phenolic substances, especially anthocyanins, catechins, and yellow and green dyes, are oxidized in the wine in contact with atmospheric oxygen. These processes are partially desirable in the production of some wines (e.g., Tokaj, Madeira, Sherry, Port, or Xerez wines), where a higher level of oxidation compounds is expected and positively evaluated [6], but these oxidations are undesirable in the production of varietal and sparkling wines.

SO_2_ is one of the most important agents that is added to wines to prevent these processes. The use of SO_2_ in winemaking is due to its ability to be an effective antioxidant, preventing the activity of the oxidases, as well as its antimicrobial property [85]. However, the addition of SO_2_ to wines can give the wine undesired flavors and aromas and can raise health-related objections due to serious allergic reactions incurred by sulfite-sensitive individuals, such as headaches, abdominal pain, and dizziness [79,92]. Ascorbic acid (vitamin C) and its optical isomer, have been widely used as an antioxidant in winemaking and are considered an alternative to sulfites, especially for white wine production, primarily because of its ability to scavenge molecular oxygen [96] and minimize the oxidative spoilage (browning) of white wine. Baroň [99,100,101] states that ascorbic acid has an immediate effect due to its high antioxidant activity. However, it protects the must or wine only against short intensive aeration, but not against long-term oxidation. Therefore, it is not effective in the long-term storage of wine. To date, a single replacement product performing the same roles as SO_2_ has not been found.

Management of oxygen is important for the browning potential of grape juice or wine. During the pressing process, exposure to oxygen was studied by Day et al. [102]. The authors stated that pressing management affects the wine composition more than handling management and it is possible to influence the concentration of particular classes of chemicals. Controlled oxygenation in the press can be used in decreasing the concentration of polyphenols. Targeted oxygenation of must or mash as a way of decreasing polyphenols in white wines without impact on sensorial changes of wines was published by Ailer et al. [98] and Pokrývková et al. [73]. If we oxidize mash or must by its exposure to atmospheric oxygen for a while without the use of sulfur dioxide or any other antioxidants, we remove excess phenolic substances from it. This means that antioxidants are not used in the grape processing technology until the must has been clarified. Phenolic substances in must or mash are oxidized with atmospheric oxygen and sedimented during sludge removal. The sulfur dioxide is used for the first time after juice clarification. In the later stages of maturation and in the final product, wine with a minimum content of phenolic substances is less prone to oxidation [73].

A protective role against browning has been attributed also to ellagitannins which may be included in the oxidation reactions of both red and white wines. The compounds ellagitannins and ellagic acid are naturally present in wood barrels and wood chips or may be used as oenological tannins. Their role in the wine oxidation process is due to their ability to quickly absorb dissolved oxygen and support the hydroperoxidation of wine components. Thus, these compounds positively influence wine color attributes and protect it from the browning process [103,104].

Oxygen harms young wine directly (oxidizes it) as well as indirectly. Aerobic conditions may help to develop unwanted microflora in the wine and irreversibly reduce its quality. There will be some oxygenation during each wine handling, but it is also the duty of the oenologist to keep it to a minimum. Polyphenols are characterized by their high antioxidant activity. This positive activity consists in scavenging free radicals and contributed to the phenomenon of the “French paradox” [105,106]. In the technology of wine production, it is necessary to take into account the beneficial aspects of phenolic substances, especially in red wine, but also their relative harmfulness in white and rosé wines. Young wine is not prone to oxidation as it still contains enough free SO_2_ from sulfurization and also residual CO_2_. However, at some point after the wine has been bottled, the sulfur dioxide is transformed into a bound form and ceases to act.

### 3.4. Corrective Solutions in Browning

Oxygen reactions with phenolics are catalyzed by oxidative enzymes to form peroxides and quinones, which further chemically oxidize ethanol to sensory negative acetaldehyde. Highly oxidized browned wine is no longer enough to treat only with sulfur dioxide. In the winemaking, several fining agents were used to decrease the level of brown compounds, mainly active charcoal and polyvinylpolypyrrolidone (PVPP). However, even though the efficiency of these agents is high, the side effect of their use is a change in the wine’s sensory characteristics, especially if they are applied at high levels [107].

The wine can be treated with activated carbon in a dose of 15–30 g per hectoliter of wine. It is a drastic intervention when, besides the polyphenols, we also remove bouquet substances and other valuable components from the wine. A gentler way to remedy oxidation is to use polyvinylpolypyrrolidone (PVPP). Its impact on changes in chemical composition and wine properties has been described by several authors [108,109].

Another possible way to remove oxidation is to re-ferment it with yeast [110], (so-called fermentation through “4”). Yeast can do a great service to an oxidized wine—they use part of the oxygen to produce sterols and long-chain fatty acids important for their growth. This will restore the wine’s reductive color and freshness.

## 4. Atypical Aging

Even maintaining all the conditions of correct vinification and using a seemingly faultless grape, sometimes the required result is not achieved. For instance, in the form of negative changes in the sensory wine profile. When this occurs, it is called “atypical wine aging”. Atypical aging (ATA) is not considered a common fault but based on its presence and contribution to the olfactory perception of wine from the view of its chemical and sensory characteristics, it is still a hot topic [111]. The causes of its occurrence can be found in the vineyard, from where the basic ingredient for wine-producing comes [13].

Atypical aging (ATA) began to emerge in the 1990s when integrated grape production expanded significantly coupled with the extensive grassland of vineyards. Its origin is attributed to the competition of green land cover for the vineyard and the associated stress [112].

Affected wines are characterized by a faulty aroma, referred to as a “Fox ton”, “wet rag”, “naphthalene”, and “bean sprouts” [113]. At the same time, these faulty aromas completely suppress the desired, fruit-floral aroma. Disharmony in the uptake of macro- and micronutrients from the soil also causes taste faults, with wine having an unpleasant and short persistence [13].

### 4.1. Sensory Attributes of ATA

#### 4.1.1. Threshold Levels

Recognition of ATA off-flavors is due to the wine complexity not being set to the threshold value of AAP concentration, however, several studies into the possible connection between the concentration of AAP and the appearance of off-odors have not confirmed a clear positive correlation [111,112,114,115]. Although in general, AAP is perceived as the main factor responsible for ATA development. Sensory recognition of ATA based on the AAP concentrations was investigated in several studies, however, a very important role is played by the wine type and its typical aroma. In general, threshold concentrations of AAP vary from 0.5 to 1.0 μg L^−1^, and in red wines, aroma alteration is observed in concentrations over 1.5 μg L^−1^ [116]. Typical off-odors can be masked by the intensive fresh and fruity wine aroma [111]. Even lower threshold concentrations (less than 0.5 μg L^−1^) have been observed in meager and light wines [112,114].

This fault is mentioned directly in the scoring system of some sensory wine competitions among the fundamental faults of wine. The organizers thus warn the assessors to take the ATA phenomenon seriously [111,117].

Olfactory perception of ATA and its connection with the AAP levels was investigated in Croatian wines by Alpeza et al. [111]. Wines with the occurrence of ATA were evaluated by the sensory analysis by the methods of “100 points” and “Yes/No” and AAP analyses were done by GC-MS. The results of AAP analyses corroborated the presence of ATA in all samples; the concentrations of AAP were 0.3–4.4 μg L^−1^. The authors stated that the occurrence of ATA may be associated with the regional, climatic conditions in a particular vintage. According to this study, the intensity of perception of off-flavors corresponding to ATA did not highly correlate with the concentration of AAP as the main chemical descriptor [111].

#### 4.1.2. Classification of the ATA Sensory Attributes

The sensory profile of the wine aroma connected with ATA perception can be divided into two groups [118,119].

The off-odors of the first group are described as chemical tones of naphthalene beads, naphthalene, soap scents, laundry detergents, furniture polishes, shoe pastes, old wax, jasmine scent, acacia blossom, lemon blossom, and dry laundry. The aroma is even more intense with the increasing content of SO_2_ [111,118,120].

The smells of wet towels, wet wool, dirty dishes, dishwashers, and dried urine are classified in the second group. These odors imply a transition to the reductive stage in sensory profiling and can make it difficult to identify the fault. In both cases, the fruit, flower, or mineral variety character disappears, partially due to acid-catalyzed ester hydrolysis and oxidation of monoterpenes and the ATA symptoms dominate [118]. Wine loses color over time and becomes lighter, its taste is thin and empty, with a typical metallic bitterness [111]. The aromas from the first group are linked with the levels of AAP in wine and it is possible to simulate them with the addition of the compound [111].

On the other hand, it is difficult to imitate the scents of the second group in wine as they are assumed to come from IAA metabolites (e.g., indole and skatole) and contribute, however; to concentrations below the threshold levels and unpleasant tones. Especially skatoles should be given higher attention due to their fecal odor [112,118,121].

### 4.2. Biochemical Background of ATA Development

The compound that has been proved to have the main responsibility for atypical aging is 2-aminoacetophenone (AAP), even at a concentration level of 1 µg L^−1^ of wine. On the other hand, even though AAP, appears as the main chemical descriptor, it is not a rule that an increase in AAP concentration results in the enhanced development of ATA [111,118].

AAP is formed from indolyl-3-acetic acid (IAA), a taste- and sensory-neutral compound, after sulfurization of young wine and L-tryptophan [111,122,123,124,125]. Biosynthesis of IAA is proposed by two main pathways in vines: the tryptophan-dependent and tryptophan-independent pathways. In the tryptophan independent pathway of IAA biosynthesis are indole or indole-3-glycerol phosphate as the main precursor [126,127].

Indole-3-acetic acid (IAA) is an essential plant auxin, occurring in all stages of plant growth and development [118].

In the tryptophan-dependent pathway, the biosynthesis is based on the conversion of tryptophan (TRP). Zhao [128] described it in a two-step conversion as shown in Figure 6.

In the first step, TRP is transformed to indole-3-pyruvate (IPA) by replacing the amino group from tryptophan and the creation of IPA with the alfa-keto-carboxylic group. The process is accompanied by the TAA family of amino transferase. In the second step, IAA is formed from IPA in the presence of the YUCCA (YUC) family of flavin-containing monooxygenases. In this step, NADPH reacts with oxygen and is catalyzed by the YUC flavin-containing monooxygenases [128].

IAA is a plant hormone that is stored in grape berries in higher concentrations as a result of plant stress. The presence of superoxide and hydroxyl radicals, which are formed just after the addition of sulfur dioxide to young wine, is usually necessary for the formation of AAP from IAA, usually during the first cellar handling.

Christoph et al. [122] and Nardin et al. [126] described the degradation of IAA (Figure 7) as follows: IAA is degraded in the fermentation process and during wine maturing. The first step is a cleavage of the pyrrole ring and formation of 3-(2-formylaminophenyl)-3-oxopropanoic acid, which is triggered by superoxide radical that originated mostly from the reaction of sulfite to sulfate [122]. A further step is the decarboxylation of 3-(2-formylaminophenyl)-3-oxopropanoic acid to produce formyl-2-aminoacetophenone and finally AAP. Alternatively, oxidized indole-acetic acid [129] can be formed. Conditions for metabolic conversion of IAA to AAP are associated with natural factors such as cooler climate, the activity of yeast, the availability of nutrients and free SO_2,_ and other factors such as ripeness, hydric stress, and irrigation [111,116,118,126]. Thus, the matrix effect seems to be more important than the initial concentration of IAA. The example provides red wines, that usually do not suffer from ATA, however, they contain approx. 10-times higher IAA levels than white wines [118]. The explanation could be the high antioxidant capacity coming from the higher concentration of phenols in red wines [126].

It is an unfortunate fact that until the young wine is sulfurized with a normal, necessary dose of sulfur dioxide, the potential for atypical aging cannot be determined with certainty. ATA does not occur in red wine because, after the first sulfurization of young wine, superoxide and hydroxyl radicals react primarily with tannins and not with indolyl-3-acetic acid.

### 4.3. Preventive Measures of ATA

The contribution of sulfites to the conversion of IAA to AAP was studied by Christoph et al. [122] and Hoenicke et al. [129]. Spiking of the model winelike solution with sulfites resulted in the disappearance of 50% of the IAA and the formation of AAP in the range from 0.5 to 20 mol %. The conversion rate was lower in the case when the wines were spiked with IAA. The authors assume the availability of free SO_2_ to be important to induce the conversion [122,129].

According to Schneider [118], degradation of IAA occurs even after a brief exposure of wine to trace oxygen concentration, thus even if in the cellar operation where the oxygen absorption is minimized, the reaction cannot be excluded. As a strong radical scavenger to prevent the formation of ATA, the use of ascorbic acid additions in white wines (similarly to the use of tannins in red wines) was confirmed to be efficient. Viticulture stress factors including drought, UV-B radiation, nutrient deficiencies, over-cropping, and premature harvest are at the very origin of a wine matrix prone to producing ATA. Enological factors play a minor role, although skin contact and yeast strain have some impact as far as they affect the presence of oxygen radical scavengers like polyphenols and yeast metabolic products [11,118].

The most important preventive measures include the optimization of the moisture regime (regulation of grass cover in vineyards, irrigation of vineyards) and the optimization of vineyard fertilization, especially with nitrogen [115,130].

No reliable analytical techniques are yet available to predict the susceptibility of wine to ATA and viticultural measures cannot confidently prevent its occurrence [123,130]. Therefore, an accelerated aging test has been proposed and put into practice [131]. This is a test where ascorbic acid is added to one of two flasks of pure wine. Both flasks are stored at 37 to 45 °C for three to four days. After cooling, both samples are evaluated for odor. If the sample without the addition of ascorbic acid shows ATA, then the wine is susceptible to its formation.

### 4.4. Corrective Solutions of ATA

Defects in nutrition can be operatively solved by applying immediately acceptable macro- and microelements by foliar application of nutrition. The effectiveness of several antioxidant adjuvants against the possible development of ATA was studied by Nardin et al. [126]. The use of ascorbic acid and grape tannin resulted in the reduced production of IAA precursors. Th eapplication of Galla tannin had a protective role especially during the storage period, as despite the IAA content, the formation of AAP was eliminated. A promising capability in the prediction of the possibility of AAP formation in wine in the process of fining was shown in ANCOVA linear modeling, using the grape variety, the IAA content before aging, and the antioxidant treatment of the must.

However, even though the literature offers partial solutions for ATA prevention, a clear method for removing AAP from wine is not confirmed. We believe that this is a clear argument that it cannot be removed without the associated costs and damages.

Clarification with fresh healthy yeast or by repeated fermentation (via “4”) can alleviate the presence of this defect due to the masking effect on the ATA off-aromas perception [118].

## 5. Conclusions

The wine faults discussed in the review are still hot topics in the winemaking industry. Consumption of faulty or sick wine does not cause a pleasant experience, it causes unpleasant feelings, and we do not ask for another sip. This results in a direct negative economic impact on the producer, but the brand image also suffers significantly. Yeast products, incorrect sulfurization regime, high content of undesirable phenolic substances, oxygen regime, and growing stress are precursors for the occurrence of defects of non-microbial origin that were the interest of this review. The latest scientific background provides findings and recommendations, particularly for the prevention of these three wine faults. The primary approach is to avoid growing stress and the use of flawless grapes. Optimal conditions for yeast and control mechanisms in winemaking are considered the best solutions to prevent the sulfur off-odors from occurring. Elimination of polyphenols as the precursors responsible for browning together with the use of antioxidants and lowering the oxygen access during winemaking create an efficient approach for browning prevention. Atypical aging prevention lies in the optimization of moisture and nutrition regime, and avoiding growth stress from excessive use of grassland cover and a fresh yeast can be used as a partial solution for ATA masking. With current knowledge, the wine faults can be reliably prevented. In the winemaking process, an essential role is played by the current analytical methods which provide important information for the early identification of potential problems. The review also provides valuable information, e.g., for producers of specific noble yeasts to eliminate the development of the faults. A further precondition for not having faulty wine in stock is the precision and consistency of the staff involved in the oenological procedures. Prevention is essential because troubleshooting is costly, difficult, and never perfect.

## Figures and Tables

**Figure 1 molecules-27-03535-f001:**
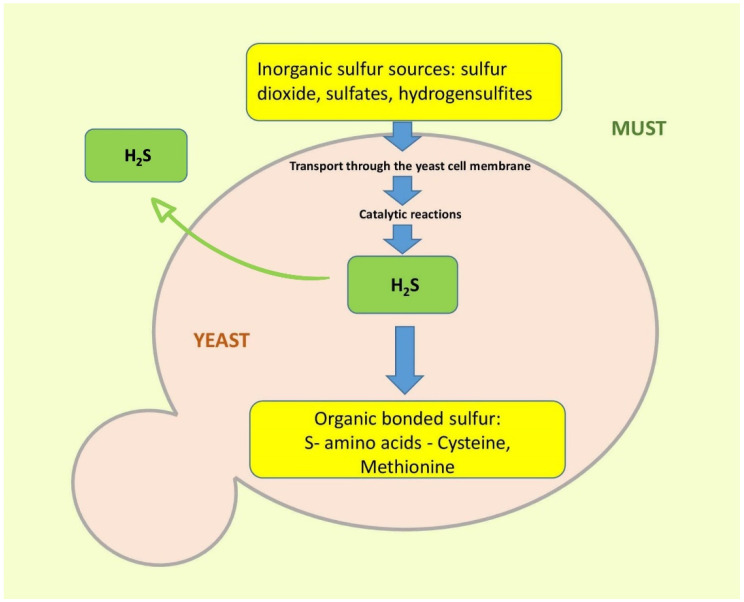
Schematic sulfur reduction pathway and formation of amino acid biosynthesis in *S. cerevisiae*, based on [34,38,40].

**Figure 2 molecules-27-03535-f002:**
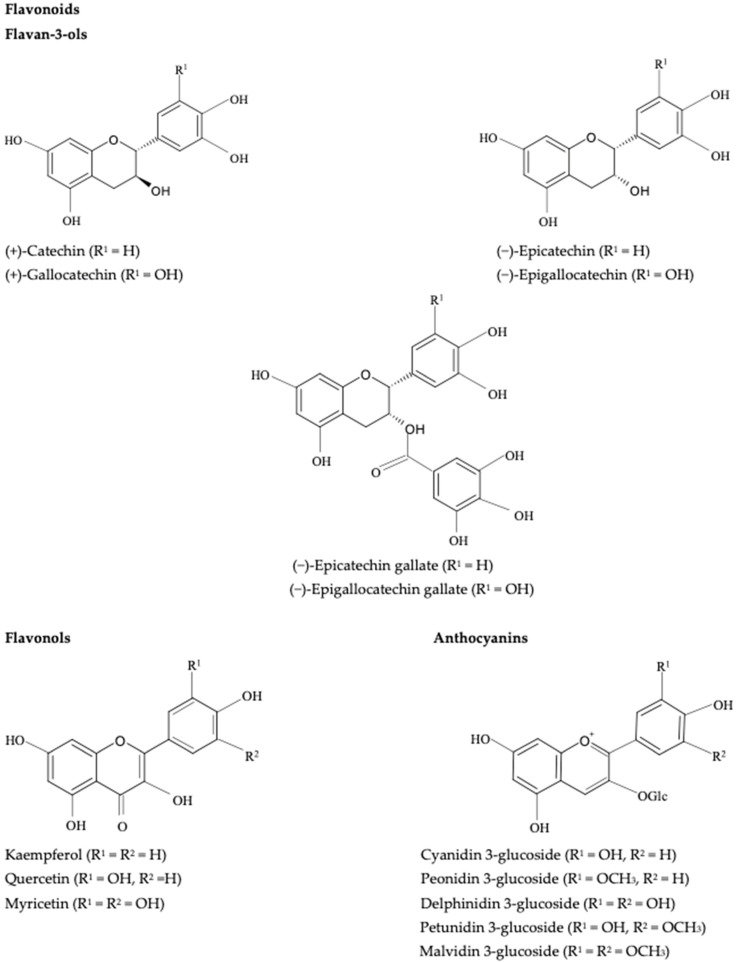
Chemical structure of the most important flavonoids in wine.

**Figure 3 molecules-27-03535-f003:**
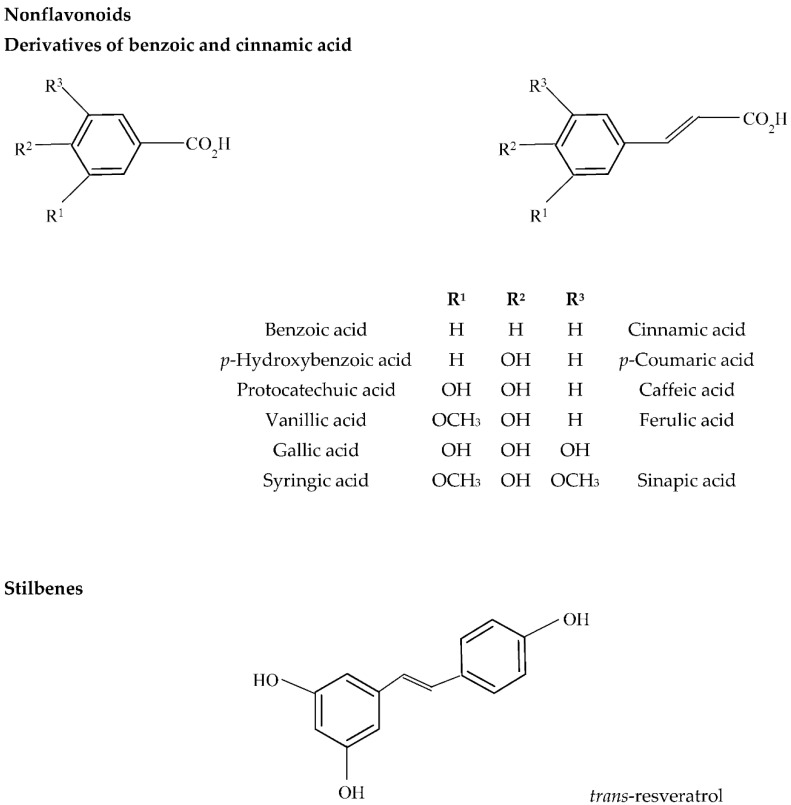
Chemical structures of the most important nonflavonoids participating in enzymatic oxidation in wine.

**Figure 5 molecules-27-03535-f005:**
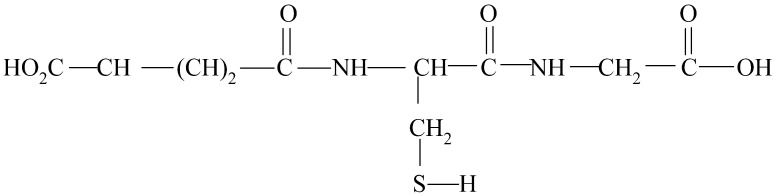
Chemical structure of glutathione.

**Figure 6 molecules-27-03535-f006:**
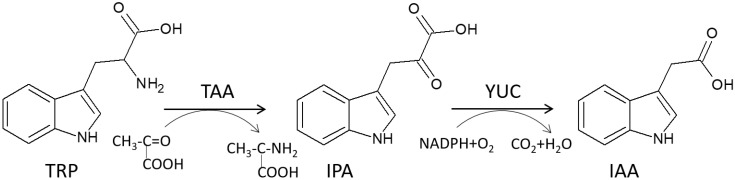
Transformation of Tryptophan to Indole-3-Acetic Acid in Plants, based on [128].

**Figure 7 molecules-27-03535-f007:**
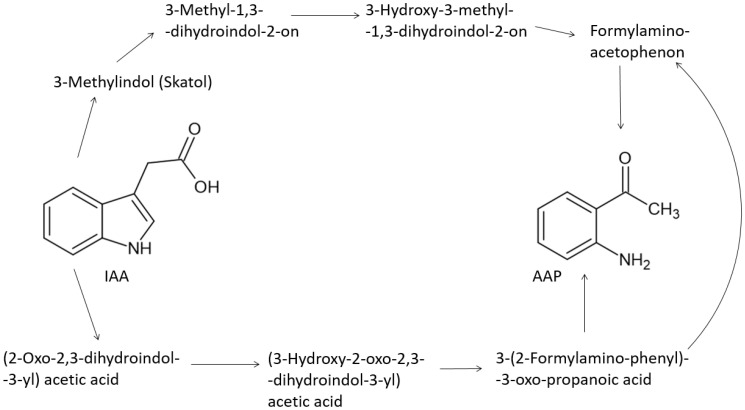
Degradation of indole-3-acetic acid to 2-aminoacetophenone, based on Christoph et al. [122] and Nardin et al. [106].

**Table 1 molecules-27-03535-t001:** Selected sulfur volatile compounds and their detection thresholds.

Compound	Aroma Description	Odor Detection Threshold (µg L^−1^)	References
Hydrogen sulfide	Rotten egg, sewage-like, vegetal	1.1–1.6	[24,26]
Methanethiol	Cooked cabbage, onion, putrefaction, rubber	1.8–3.1	[24,27]
Ethanethiol	Onion, rubber, natural gas, faecal, earthy	1.1	[27,28]
Dimethylsulfide	Asparagus, corn, molasses, boiled cabbage, canned corn, blackcurrant, truffle	25	[28]
Diethylsulfide	Cooked vegetables, onion, garlic, rubber	0.90	[27,28]
Dimethyl disulfide	Cooked cabbage, intense onion	29	[27,28]
Diethyl disulfide	Onion, garlic, burnt rubber	4.3	[27,28]
3-(methylthio)-1-propanol (methionol)	Cauliflower, cabbage, potato	500	[29]

## Data Availability

Not applicable.

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
