# Peer review of "Wine Faults: State of Knowledge in Reductive Aromas, Oxidation and Atypical Aging, Prevention, and Correction Methods"

_molecules, 2022, doi:10.3390/molecules27113535_

Round 1

Reviewer 1 Report

The manuscript is prepared professionally. It includes a well-crafted abstract and an exhaustive introduction that justifies the research undertaken. The introduction points to the deficiencies in the literature on the subject. The aim is clearly defined.  The discussion of the results is well prepared. The conclusions are well-defined. The illustrative material is appropriate.

In my opinion, the manuscript after corrections, will be suitable for publication in a journal.

Detailed comments:

1-Below sentence needs 3-4 updated references

We are currently witnessing a sensory wine revolution. There are liberal styles and fractions, where oxidation, turbidity or excessive content of phenolic substances are accepted even in white wine. Protein-dependent turbidity and crystalline sediments do not need to be classified as wine faults in the current liberal conditions. If there is a market for such wines, and they are produced according to clear applicable rules, it is necessary to respect them. Despite this liberal era, the two-thousand-year history has set and shaped certain rules. It is documented and published in professional and scientific literature, what can be considered as a faulty wine with the impact on wine sensory profile. Consuming the faulty wine does not bring a pleasure to the consumer, it causes unpleasant feelings, and the consumer does not ask for another sip. 

Prokes, K.; Baron, M.; Mlcek, J.; Jurikova, T.; Adamkova, A.; Ercisli, S.; Sochor, J. The Influence of Traditional and Immobilized Yeast on the Amino-Acid Content of Sparkling Wine. Fermentation 2022

......

Sulfur as an important element in biological systems is metabolized in various compounds, responsible for desired but also some unpleasant aromas. Volatile sulfur compounds are produced by microorganisms in sulfate assimilatory and dissimilatory reduction pathways [1].

Needs more references

Author Response

Author's Reply to the Review Report (Reviewer 1)

We would like to thank Reviewer 1 for his/her comments and suggestions for manuscript improvement. We tried to correct the manuscript according to them.

The manuscript is prepared professionally. It includes a well-crafted abstract and an exhaustive introduction that justifies the research undertaken. The introduction points to the deficiencies in the literature on the subject. The aim is clearly defined.  The discussion of the results is well prepared. The conclusions are well-defined. The illustrative material is appropriate.

In my opinion, the manuscript after corrections, will be suitable for publication in a journal.

Detailed comments:

1-Below sentence needs 3-4 updated references

We are currently witnessing a sensory wine revolution. There are liberal styles and fractions, where oxidation, turbidity or excessive content of phenolic substances are accepted even in white wine. Protein-dependent turbidity and crystalline sediments do not need to be classified as wine faults in the current liberal conditions. If there is a market for such wines, and they are produced according to clear applicable rules, it is necessary to respect them. Despite this liberal era, the two-thousand-year history has set and shaped certain rules. It is documented and published in professional and scientific literature, what can be considered as a faulty wine with the impact on wine sensory profile. Consuming the faulty wine does not bring a pleasure to the consumer, it causes unpleasant feelings, and the consumer does not ask for another sip. 

Answer: We thank the reviewer for his/her comment and we tried to correct the manuscript according to them. All introduction part was written in own words from long-term experiences, knowledge, and perception of the topic, but we revised the manuscript and added appropriate references, supporting the statements in the introduction as well as in other parts of the article. The new references are indicated in yellow color in the text of the manuscript.

Prokes, K.; Baron, M.; Mlcek, J.; Jurikova, T.; Adamkova, A.; Ercisli, S.; Sochor, J. The Influence of Traditional and Immobilized Yeast on the Amino-Acid Content of Sparkling Wine. Fermentation 2022

Answer: We embedded the reference in an appropriate place in relation to the decomposition of amino acids to sulfur compounds during longer maturation – line 181, reference no.47.

......

Sulfur as an important element in biological systems is metabolized in various compounds, responsible for desired but also some unpleasant aromas. Volatile sulfur compounds are produced by microorganisms in sulfate assimilatory and dissimilatory reduction pathways [1].

Needs more references

Answer: Thank you for the comment, we added 22 new references in the text of the manuscript to support information. All references were reordered in the Zotero reference manager.

Reviewer 2 Report

The objective of this review was to contribute to the current knowledge of common wine faults related to reductive aromas, browning, and atypical aging and pay attention to them in terms of sensory attributes, chemical background and preventive and corrective measures in wine during the fermentation and after bottling.

The subject is very important for wine production. The review provides an update and deepening knowledge of this subject. The bibliography consulted is numerous and accurately, considering the different aspects involved in the subject.

Particularly relevant are the consideration of preventive measures of formation and the corrective solutions in reductive aromas.

The same for preventive measures of wine browning and corrective solutions in browning, and preventive measures and corrective solutions of atypical aging.

Only minor aspects should be reviewed.

In Table 1, in the aroma description of dimethylsulfide the authors cited asparagus twice.

In the same Table says Referencies  as title of last column.

In line 150 the authors indicated wineyards, instead of vineyards.

Author Response

Author's Reply to the Review Report (Reviewer 2)

We would like to thank Reviewer 2 for his/her comments and suggestions for manuscript improvement. We tried to correct the manuscript according to them.

The objective of this review was to contribute to the current knowledge of common wine faults related to reductive aromas, browning, and atypical aging and pay attention to them in terms of sensory attributes, chemical background and preventive and corrective measures in wine during the fermentation and after bottling.

The subject is very important for wine production. The review provides an update and deepens knowledge of this subject. The bibliography consulted is numerous and accurate, considering the different aspects involved in the subject.

Particularly relevant are the consideration of preventive measures of formation and the corrective solutions in reductive aromas.

The same for preventive measures of wine browning and corrective solutions in browning, and preventive measures and corrective solutions of atypical aging.

Answer: We thank reviewer 2 for his/her comments and we did corrections according to them.

Only minor aspects should be reviewed.

In Table 1, in the aroma description of dimethylsulfide the authors cited asparagus twice.

Answer: The duplicity was deleted.

In the same Table says Referencies  as title of last column.

Answer: We corrected the word „References“ in the last column.

In line 150 the authors indicated wineyards, instead of vineyards.

Answer: Typing mistake was corrected.

Reviewer 3 Report

This review covers some aspects of sensory wine faults, including preventive and corrective measures. However, there are some significant concerns that the authors need to address.

  • Authors should consider rephrasing the manuscript title to be more concise.
  • The Abstract should be prepared to cover all aspects of the review without unnecessary explanations, including mentioning the author’s previous research.
  • The Introduction section, which is somewhat longer than the rest of the article, is without any reference.
  • What is the purpose of Figures 3, 4, and 5?
  • What is added value of this review in the context of existing evidence?
  • Conclusions and future directions should be drawn from the literature data summarized in the review.

Author Response

Author's Reply to the Review Report (Reviewer 3)

We would like to thank Reviewer 3 for his/her comments and suggestions for manuscript improvement. We tried to correct the manuscript according to them.

  • This review covers some aspects of sensory wine faults, including preventive and corrective measures. However, there are some significant concerns that the authors need to address.

Answer: We thank the reviewer for his/her comments and questions. We tried to answer them and corrected the relevant parts of the manuscript.

  • Authors should consider rephrasing the manuscript title to be more concise.

Answer: We proposed new shorter title „Wine faults: state of knowledge in reductive aromas, oxidation, and atypical aging, prevention, and correction methods“ – see changes in the manuscript.

  • The Abstract should be prepared to cover all aspects of the review without unnecessary explanations, including mentioning the author’s previous research.

Answer: We rewrote the abstract, and omitted previous research. The changes are marked in red color.

  • The Introduction section, which is somewhat longer than the rest of the article, is without any reference.

Answer: All introduction part was written in own words from long-term experiences and knowledge, but we revised the manuscript and added appropriate references, supporting the statements in the introduction as well as in other parts of the article. The newly added references are indicated in yellow color in the text of the manuscript.

  • What is the purpose of Figures 3, 4, and 5?

Answer: We thank the reviewer for the question. Even the structure of the phenolic compounds is known, we consider that it would be good and helpful to have formulas of the most important phenolic compounds in wines, flavonoids and nonflavonoids (Fig 2 and 3), especially nonflavonoids which are easily oxidized compounds and influence on changing the color to brown in white wines. Moreover, Fig 3 and 4 present basic and important compounds participating in enzymatic oxidation.

  • What is added value of this review in the context of existing evidence?

Answer: We find the subject to be very important for wine production, the wine faults discussed in the review are still hot topics in the winemaking industry. We tried to provide the current state of knowledge, with basic biochemical background and sensorial impact of the changes on wine quality. We tried to discuss the important aspects of the subject with particular consideration of relevant preventive measures for the formation of these faults and their corrective solutions.

And this is also the benefit of the article: To improve the state of knowledge of these three faults for practice and further development of the scientific discipline. Focus on prevention, not only on faults correction. Summarize the state of analytical methods for the determination of wine error precursors. The summary will bring valuable information e.g. for producers of specific noble yeasts, selected for minimal production of AAP, H2S, and other responsible compounds. The review supports the current state of knowledge with the potential for managing the faults during winemaking and storage.

  • Conclusions and future directions should be drawn from the literature data summarized in the review.

Answer: We tried to rewrite the conclusion by emphasizing the importance for winemakers and by indicating the future directions. The changes are marked by red color in the text.

Round 2

Reviewer 3 Report

The authors have made additional efforts to improve the manuscript.

References in the Introduction added. However, of the total 13, reference 4 appears in all paragraphs.

Regarding Figures 3 and 4, please add in their titles that chemical structures present compounds participating in enzymatic oxidation.

Author Response

Author's Reply to the Review Report (Reviewer 3)

We would like to thank Reviewer 3 for his/her comments and suggestions for manuscript improvement. We tried to correct the manuscript according to them.

  • The authors have made additional efforts to improve the manuscript.
  • References in the Introduction added. However, of the total 13, reference 4 appears in all paragraphs.

Answer: We thank the reviewer for his/her comments. We added together 22 new references in the first round of manuscript correction, and 13 were embedded in the introduction part. Reference no. 4 is the author's book, which was previously written, and besides the other important topics about Winemaking and Sommeliership, wine faults were also discussed, therefore the reference is embedded in the introduction as the ideas are mentioned. If there are any other issues that need to be corrected/improved, we would like to ask the reviewer what he/she specifically considers necessary to be corrected.

  • Regarding Figures 3 and 4, please add in their titles that chemical structures present compounds participating in enzymatic oxidation.

Answer: We thank the reviewer for this comment, we corrected the titles of Figures 3 and 4, see red changes in Lines 417 and 474.
